# (*E*)-Piplartine Isolated from *Piper pseudoarboreum*, a Lead Compound against *Leishmaniasis*

**DOI:** 10.3390/foods9091250

**Published:** 2020-09-07

**Authors:** Juan C. Ticona, Pablo Bilbao-Ramos, Ninoska Flores, M. Auxiliadora Dea-Ayuela, Francisco Bolás-Fernández, Ignacio A. Jiménez, Isabel L. Bazzocchi

**Affiliations:** 1Instituto Universitario de Bio-Orgánica Antonio González and Departamento de Química Orgánica, Universidad de La Laguna, Avenida Francisco Sánchez 2, 38206 La Laguna, Tenerife, Spain; biojuancarlos@hotmail.com; 2Instituto de Investigaciones Fármaco Bioquímicas, Facultad de Ciencias Farmacéuticas y Bioquímicas, Universidad Mayor de San Andrés, Avenida Saavedra 2224, Miraflores, La Paz, Bolivia; eflores5umsa@gmail.com; 3Departamento de Parasitología, Facultad de Farmacia, Universidad Complutense de Madrid, Plaza Ramón y Cajal s/n, 28040 Madrid, Spain; pablobil15@yahoo.com (P.B.-R.); mda_3000@yahoo.es (M.A.D.-A.); francisb@ucm.es (F.B.-F.); 4Laboratorio de Parasitología y Entomología, Instituto Nacional de Laboratorios de Salud (INLASA), Pasaje Rafael Zubieta 1889, Zona Miraflores, La Paz, Bolivia; 5Departamento de Farmacia, Universidad CEU-Cardenal Herrera, Avenida Seminario s/n, 46113 Moncada, Valencia, Spain

**Keywords:** *Piper pseudoarboreum*, bioassay-guided fractionation, leishmanicidal activity, alkamides, (*E*)-piplartine

## Abstract

The current therapies of leishmaniasis, the second most widespread neglected tropical disease, have limited effectiveness and toxic side effects. In this regard, natural products play an important role in overcoming the current need for new leishmanicidal agents. The present study reports a bioassay-guided fractionation of the ethanolic extract of leaves of *Piper pseudoarboreum* against four species of *Leishmania* spp. promastigote forms, which afforded six known alkamides (**1**–**6**). Their structures were established on the basis of spectroscopic and spectrometric analysis. Compounds **2** and **3** were identified as the most promising ones, displaying higher potency against *Leishmania* spp. promastigotes (IC_50_ values ranging from 1.6 to 3.8 µM) and amastigotes of *L. amazonensis* (IC_50_ values ranging from 8.2 to 9.1 µM) than the reference drug, miltefosine. The efficacy of (*E*)-piplartine (**3**) against *L. amazonensis* infection in an in vivo model for cutaneous leishmaniasis was evidenced by a significant reduction of the lesion size footpad and spleen parasite burden, similar to those of glucantime used as the reference drug. This study reinforces the therapeutic potential of (*E*)-piplartine as a promising lead compound against neglected infectious diseases caused by *Leishmania* parasites.

## 1. Introduction

Leishmaniases are neglected tropical diseases caused by the infection with *Leishmania* parasites, and are transmitted by the bite of a sand fly belonging to the genera *Lutzomyia* and *Phlebotomus*. Leishmaniases are endemic in large areas of the tropics, subtropics, and the Mediterranean basin, and are among the major neglected tropical diseases causing morbidity worldwide. Recently, it has broken out of its traditional boundaries and has been reported in new geographic locations with atypical disease manifestations involving novel parasite variants. Cutaneous leishmaniasis (CL) is endemic in more than 70 countries, with an estimated annual incidence of 1.5–2 million new cases, and clinical manifestations ranging from small skin nodules to massive destruction of the mucous tissues. CL is mainly caused by *Leishmania major* in the Old World and by *L. amazonensis* and *L. braziliensis* in the New World, specifically in Brazil [1]. In spite of the high prevalence, and advances in the chemotherapy for leishmaniasis, the current available drugs, including pentavalent antimonials, amphotericin B, miltefosine, paromomycin, and pentamidine are compromised by the emergence of resistance, variable sensitivity between species, adverse side effects, requirements for long courses of administration, and high cost [2]. These drawbacks and the absence of vaccines underline the urgent need for searching alternative treatments with acceptable efficacy and safety profile.

Natural products are an important source of leishmanicidal drugs owing to their accessibility, structural diversity, low cost, and possible rapid biodegradation [3,4,5]. In South America, where resorting to medicinal plants represents a primary health care measure of the native population, several species of *Piper* genus are widely used as a remedy to relieve the symptoms of leishmaniasis disease. Thus, the leaves of *Piper aduncum*, *P. loretoanum,* and *P. hispidum* are used as poultices for healing wounds and to treat the symptoms of CL [6,7]. In addition, *Piper* species are used as culinary spices, and as a food preservative to control food spoilage and pathogenic microorganisms. In particular, *P. nigrum* (black pepper) is worldwide popular as a flavoring for food [8]. Phytochemical investigations of *Piper* species have reported numerous metabolites with ecological and medicinal properties, including amides, pyrones, lignanes, terpenes, and flavonoids [8]. Alkamides, also named piperamides, are characteristic bioactive constituents in *Piper* species [9]. In particular, (*E*)-piplartine, also called piperlongumine, is the major natural alkaloid from *P. longum* and *P. tuberculatum*, and in vitro and in vivo studies have demonstrated its promising pharmacological properties such as antioxidant, anxiolytic, anti-atherosclerosis, antidiabetic, and antiparasitic against neglected tropical diseases [10]. Moreover, (*E*)-piplartine is reported to kill a large variety of cancer cells while remaining nontoxic to normal cells, highlighting its therapeutic potential [11,12].

In previous investigations, we reported the isolation of an unprecedented chlorine-containing piperamide along with several known compounds and their antileihmanicidal activity from *Piper pseudoarboreum* [13]. In continuous research toward the discovery of natural occurring leihmanicidal agents, we report herein on the isolation and structure elucidation of six known alkamides from the leaves of *P. pseudoarboreum* Yunker through a bioassay-guided fractionation carried out against four promastigote strains of *Leishmania*. Compounds **2** and **3** were further evaluated on intracellular amastigotes of *L amazonensis* and *L. infantum*. (*E*)-piplartine (**3**) was selected to be assayed in an in vivo model for cutaneous leishmaniasis.

## 2. Materials and Methods

### 2.1. General Experimental Procedures

The structure of the isolated compounds were elucidated using spectrometric and spectroscopic methods, and comparison with data previously reported. The Nuclear Magnetic Resonance (NMR) experiments were recorded on Bruker Avance 400 and 500 spectrometers (Bruker Co. Billerica, MA, USA); chemical shifts were referred to the residual solvent signal (CDCl_3_: δ_H_ 7.26, δ_C_ 77.36) (acetone *d*_6_: δ_H_ 2.09, δ_C_ 30.60 and 205.87), using trimethylsilane (TMS) as internal standard. Electron Impact Mass Spectrometry (EIMS) and High Resolution Electron Impact Mass Spectrometry (HREIMS) were recorded on a Micromass Autospec spectrometer (Micromass, Manchester, UK). Silica gel 60 (15–40 mm) and silica gel 60 F254 for column chromatography and Thin Layer Chromatography (TLC), respectively, were purchased from Panreac (Barcelona, Spain). Sephadex LH-20 was obtained from Pharmacia Biotech (Pharmacia, Uppsala, Sweden). Centrifugal planar chromatography was carried out in a Chromatotron instrument (model 7924T, Harrison Research Inc., Palo Alto, CA, USA) on manually coated silica gel 60 GF_254_ (Merck, Darmstadt, Germany) using 4-mm plates. The spots were visualized by UV light and heating silica gel plates sprayed with H_2_O-H_2_SO_4_-AcOH (1:4:20).

### 2.2. Chemicals and Reagents

All solvents used were of analytical grade and purchased from Panreac (Barcelona, Spain). (*E*)-Piplartine, Scheneider’s insect medium, RPMI-1640, fetal bovine serum (FBS), 4-(2-hydroxyethyl)-1-piperazineethanesulfonic acid (HEPES), resazurin sodium salt, and sodium dodecyl sulphate (Sigma-Aldrich, St Louis, MO, USA), L-glutamine (Avantor Performance Material Inc., PA, USA), trypsin (Merck, Darmstadt, Germany), penicillin Penilevel^®^ 100.000 U.I. (ERN laboratories, Barcelona, Spain), streptomycin sulphate (Reig Jofré laboratories, Barcelona, Spain), and Glucantime^®^ (Merial Laboratories, Barcelona, Spain).

### 2.3. Plant Material

Leaves of *Piper pseudoarboreum* Yunck. were collected in November 2009 at Iquitos, Maynas Province, Department of Loreto, Perú. The plant material was identified by the botanist Juan Celedonio Ruiz Macedo, and a voucher specimen (AMZ 11114) was deposited at the Amazonense Herbarium of the Universidad Nacional de la Amazonia Peruana, Iquitos, Perú.

### 2.4. Extraction, Bioassay-Guided Fractionation and Isolation

The dried leaves of *P. pseudoarboreum* (200.3 g) were powdered and extracted in a Soxhlet apparatus with 5 L of 96% ethanol. The solvent was evaporated to give 42.9 g (21.4%) of extract. The ethanolic extract (EtOH) was partitioned into dichlorometane (DCM), ethyl acetate (EtOAc), and water (H_2_O). After removing the organic solvents under reduced pressure, the DCM (9.2 g, 4.6%) and EtOAc (1.2 g, 0.6%) fractions were obtained, whereas the aqueous-soluble extract was lyophilized providing the H_2_O fraction (8.9 g, 4.5%). The most active organic fraction (DCM, 9.2 g) was chromatographed over silica gel column eluting with mixtures of hexanes-EtOAc (10:0 to 0:10, 1 L each one) to obtain seven sub-fractions (F1–F7). The most active fraction, F6 (1.5 g), was subjected to column chromatography over Sephadex LH-20 by isocratic elution (MeOH-CHCl_3_, 1:1) to afford fifteen sub-fractions, which were combined based on their TLC profiles (F6A to F6F). Preliminary nuclear magnetic resonance (NMR) studies revealed that sub-fraction F6B was rich in aromatic alkamides, and were further investigated. Thus, F6B (448.1 mg) was chromatographed by centrifugal planar chromatography on 4-mm silica gel plates, using mixtures of hexanes-EtOAc (60:40 to 50:40) as eluent to give eleven sub-fractions (F6B1 to F6B11). Sub-fraction F6B2 (21.5 mg) was further purified on silica gel by preparative TLC (3 × development, hexanes-2-propanol, 8:2) to give compounds **1** (1.7 mg) and **5** (1.4 mg). Purification of sub-fraction F6B4 (18.3 mg) by preparative TLC (2 × development, CH_2_Cl_2_-Et_2_O, 95:5) yielded compounds **3** (11.4 mg) and **4** (2.2 mg), whereas sub-fraction F6B7 (23.8 mg) gave compounds **2** (19.8 mg) and **6** (0.9 mg) after purification by preparative TLC (2 × development, hexanes-2-propanol, 8:2). The compounds were identified by NMR spectroscopy and comparison with data reported in the literature.

### 2.5. Biological Studies

#### 2.5.1. Parasites

Autochthonous isolates of *Leishmania infantum* (MCAN/ES/92/BCN83) were obtained from an asymptomatic dog from the Priorat region (Catalunya, Spain), and kindly provided by Prof. Montserrat Portús (University of Barcelona). *L. braziliensis* (2903), *L. amazonensis* (MHOM/Br/79/Maria) and *L. guyanensis* (141/93) were kindly given by Prof. Alfredo Toraño (Instituto de Salud Carlos III, Madrid).

#### 2.5.2. Cells

J774 murine macrophages were grown and maintained in RPMI-1640 medium supplemented with 10% heat-inactivated FBS, penicillin G (100 U/mL), and streptomycin (100 μg/mL) at 37 °C and 5% CO_2_ air atmosphere.

#### 2.5.3. Animals

Male BALB/c mice of 20–25 g body weigh were purchased from Harlan Interfauna Ibérica (Barcelona, Spain). All rodents were housed in plastic cages in a 12 h dark–light cycle under controlled temperature (25 °C) and humidity (70%) conditions. During the study, animals had unrestricted access to food and water.

#### 2.5.4. In Vitro Promastigotes Susceptibility Assay

In vitro antileishmanial assay was performed using a method described elsewhere [14]. Briefly, promastigotes were grown in vitro in a Schneider’s insect medium supplemented with 20% heat-inactivated FBS, penicillin (100 U/mL) and streptomycin (100 μg/mL) at 26 °C in 25 mL in tissue culture flasks, and were cultured in 96-well plastic plates (2.5 × 10^5^ parasites/well). Compounds dissolved in dimethylsulfoxide 1% (DMSO) at the suitable concentration to be tested in serial dilutions (a first screening using 100 μg/mL, and then 100, 50, 25, 12.5, 6.25, 3.12, 1.56 and 0.78 μg/mL) to get a final volume of 200 μL were added to each well. After an incubation of 48 h at 26 °C, 20 μL of 2.5 mM resazurin solution was added. Plates were then analyzed by fluorescence emission (535_ex_–590_em_ nm) using a fluorometer Infinite 200 (Tecan i-Control, Tecan Group Ltd, Männedorf, Switzerland). All tests were carried out in triplicate, and miltefosine was used as the reference drug. The antileishmanial activity of each compound was estimated by calculating the GI% (percentage of growth inhibition) and then the IC_50_ value (concentration of the compound that produced a 50% reduction in parasites).

#### 2.5.5. Cytotoxicity Assay

The cytotoxicity assay of the tested compounds was performed according to a previously described method [14]. Briefly, J774 macrophages (5 × 10^4^ cells/well) were placed in 96-well flat-bottom plates with 100 μL of RPMI-1640 medium, and allowed to attach at 37 °C and 5% CO_2_ for 2 h. Afterwards, 100 μL of RPMI-1640 medium containing the test compound in varying concentrations (100, 50, 25, 12.5, 6.25, 3.12, 1.56, and 0.78 µg/mL) were added to the cells and incubated for another 48 h. Growth controls and signal-to-noise were included. Following the aforementioned incubation time, 20 μL of 2.5 mM resazurin solution in PBS was added, and the plates were placed again in the incubator for another 3 h to evaluate cell viability. The ability of cells to reduce resazurin was determined by fluorometry as in the promastigote assay. Each concentration was assayed in triplicate. Cytotoxicity was expressed as the 50% reduction of cell viability of treated culture cells with respect to untreated culture (CC_50_).

#### 2.5.6. In Vitro Amastigote Assay

The effectiveness against intracellular amastigotes was evaluated using a fluorometric method described elsewhere [15]. Briefly, macrophages (5 × 10^4^ cells) and stationary *Leishmania* promastigotes in a ratio of 1:10 (macrophage/parasite) were seeded in each well of a microtiter plate, suspended in 200 μL of culture medium and incubated at 33 °C and 5% CO_2_ for 24 h. After this incubation time, the temperature was increased up to 37 °C for another 24 h. Cells were washed with medium several times in order to remove free non-infective promastigotes, and the supernatant was replaced by 200 μL/well of culture medium containing two-fold serial dilutions of the test compounds (ranging from 5 to 0.038 μg/mL) and the reference drug (ranging from 50 to 0.38 μg/mL). The culture medium was removed carefully to be replaced by 200 μL/well of the lysis solution (RPMI-1640 with 0.048% HEPES and 0.006% sodium dodecyl sulfate (SDS)) and incubated at room temperature for 20 min. Thereafter, the plates were centrifuged at 3500× *g* for 5 min and the lysis solution was replaced by 200 μL/well of Schneider’s insect medium. The culture plates were incubated at 26 °C for another 3 days for the transformation of viable promastigotes into amastigotes. Afterwards, 20 μL/well of 2.5 mM resazurin was added and incubated for 3 h. Plates were analyzed by fluorescence emission, and IC_50_ was determined as described above. All tests were carried out in triplicate. Miltefosine was used as reference drug and was evaluated at the same conditions.

#### 2.5.7. In Vivo Experiments

BALB/c mice were infected subcutaneously at the left hand-foot with 1 × 10^7^ promastigotes of *L. amazonensis* on day 0. Right hind paw was used as a negative control (no infection). Thirty five days after infection, chronic cutaneous leishmaniasis was developed, and animals were randomly divided into three groups (*n* = 8/group): animals treated with (*E*)-piplartine received in the foot lesions (intralesion) doses of 25 mg/kg/day for 4 days in a 15 µL volume of phosphate saline dilution/propylene glycol (9:1), a group treated with glucantime receiving 25 mg/kg/day for 4 days by intraperitoneal route, and the control group. The measurement of cutaneous lesion was monitored at 0, 35, 50, and 100 days post-infection, using a Vernier calliper to measure footpad size. The number of viable *L. amazonensis* parasites in the spleen of the different groups of mice was estimated using the limiting dilution assay method at the end of the experiment (day 100 post-infection) [16]. Mice were sacrificed, and the spleen were aseptically removed, weighed, and homogenized in Schneider’s medium supplemented with 10% FBS. Briefly, serial dilutions were prepared and distributed to 96-well microtiter plates under sterile conditions, and incubated at 26 °C. On day 7 post-incubation, wells were analyzed using an inverted microscope. The number of parasites per milligram of tissue was determined based on the tissue weight and the parasite load from the culture dilutions [17].

#### 2.5.8. Ethical Consideration

All animals were handled according to the European Union legislation Directive 2010/63/EU and Spanish law Real Decreto 53/2013 on the protection of animals used for scientific purposes. The experimental protocols involving the use of animals were approved by the local ethical committee of the University Complutense of Madrid (CEXAN170415) http://147.96.70.122/Web/Actas/CEXAN170415.pdf.

#### 2.5.9. Statistical Analysis

For in vitro assays, the antileishmanial activity (IC_50_) and cytotoxic activity (CC_50_) of compounds were analyzed by Probit test, using SPSS v20.0 software. All results were expressed as means ± standard error of the mean (S.E.M). For in vivo assays, results were analyzed by Shapiro-Wilk’s normality test, and then by one-way ANOVA with Tukey’s HSD post-hoc test. Significant differences were considered at *p-*value < 0.05, using SPSS v20.0 and Microsoft Excel 2010 software.

## 3. Results and Discussion

The ethanolic extract of the leaves of *P. pseudoarboreum* was evaluated against promastigote forms of *L. amazonensis*, *L. braziliensis*, *L. guyanensis,* and *L. infantum*. The active EtOH crude extract was further fractionated by liquid–liquid partition to obtain DCM, EtOAc, and H_2_O fractions, which were assayed for their in vitro activity against the four *Leishmania* strains.

The DCM fraction showed an improved profile compared to the crude extract, displaying IC_50_ values ranging from 14.7 to 19.1 µg/mL for the four *Leishmania* strains assayed, whereas the EtOAc and H_2_O fractions showed to be inactive (IC_50_ > 50 µg/mL). Thus, DCM fraction was further fractionated to yield seven sub-fractions. Sub-fractions F1–F4 showed to be inactive (IC_50_ > 100 µM), whereas sub-fraction F5 showed some degree of activity on the four *Leishmania* strains (IC_50_ 15.7–20.8 µg/mL) and F7 exhibited only slight potency on *L. amazonensis* and *L. brazilensis.* Moreover, the most active sub-fraction F6 exhibited higher potency than miltefosine, used as the reference drug (IC_50_ ranging from 2.2 to 3.4 µM vs. 17.7 to 30.7 µM), although showed a slightly low selectivity index taking J774 macrophages as reference mammalian cells (CC_50_ values ranging from 1.9 to 3.0 vs. 4.4 to 7.7) (Table 1).

Therefore, sub-fraction F6 was submitted to multiple chromatographic steps on silica gel and Sephadex LH-20 affording the known alkamides **1**–**6** (Figure 1). Their chemical structures were elucidated on the basis of their spectroscopic data (Appendix A) and comparison with data reported in the literature. Thus, the isolated metabolites were identified as sintenpyridone (**1**) [18], (*E*)-demethoxypiplartine (**2**) [19], (*E*)-piplartine (also known as piperlongumine, **3**) [19], (*Z*)-piplartine (**4**) [20], 3,4-epoxy-8,9-dihydropiplartine (**5**) [21], and 10,11-dihydropiperine (**6**) [20].

Alkamides **1**–**6** were tested by in vitro assays against the four strains of *Leishmania* promastigotes. The results indicated that alkamides **2** and **3** were 4.7 to 18-fold more potent (IC_50_ ranging from 1.6 to 3.8 µM) than miltefosine (IC_50_ ranging from 17.7 to 30.7 µM), and exhibited a selectivity index ranging from 3.0 to 6.6 for all *Leishmania* strains tested (Table 1). Recently, Araújo-Vilges and co-workers reported that (*E*)-piplartine was able to reduce the growth of *L. amazonensis* promastigotes (MHOM/BR/pH8) in a dose-dependent pattern, exhibiting an IC_50_ value of 179.0 µg/mL [22]. On the other hand, Capello et al. [23] reported that no antileishmanial activity on macrophages infected with *L. (L.) amazonensis* was found for (*E*)-piplartine at 50 µg/mL. We assume that such differences in potency depend to a great extent on the infecting *Leishmania* strain used in the assay and cell culture procedures.

Regarding the influence of the substitution pattern in the alkamide scaffold on the leishmanicidal activity, it seems that α,β-unsaturated carbonyl groups in both, the acyl chain and the lactam ring are critical functionalities for the activity (**2**, **3**, and **4** vs. **1**, **5**, and **6**). Moreover, isomerization of the unsaturated acyl chain leads to slight changes in the activity (**3** vs. **4**). No straightforward conclusion can be drawn from the type of functional group on the aromatic ring.

Based on the in vitro results on promastigote forms, alkamides **2** and **3** were selected to be evaluated on intracellular amastigotes of *L. amazonensis* and *L. infantum.* The results revealed that compounds **2** and **3** exhibited some degree of activity, showing two-fold higher potency on *L. amazonesis* (IC_50_ 9.1 and 8.2 µM, respectively) than on *L. infantum* (IC_50_ 17.1 and 16.1 µM, respectively). Moreover, both compounds exhibited higher activity than miltefosine on both assayed *Leishmania* strains. Furthermore, these compounds were 5- to 6-fold more potent than miltefosine on *L. amazonensis* (Table 2).

Taking into consideration its potency and efficacy on *Leishmania* promastigote and amastigote forms, and although a poor selectivity index, (*E*)-piplartine was selected for in vivo assays to investigate its potential as a lead compound targeting CL since previous toxicological studies indicate a good safety profile in murine models [10]. Previous works report the in vitro evaluation of (*E*)-piplartine against *Leishmania* spp. promastigotes [22,24] and *L. amazonensis* intracellular amastigote forms [25] as well as an in vivo study against *L. donovani* in a hamster model of visceral leishmaniasis [24]. However, to our knowledge, in vivo studies on CL have not been reported.

The in vivo assay in BALB/c mice infected with *L. amazonensis* for CL was performed by treatment of three randomly separated mice groups (8 mice per group). Thirty five days after infection, the treated mice group with (*E*)-piplartine received in the foot lesions (intralesion) a dose of 25 mg/kg/day for 4 days, whereas the treated group with glucantime received by intraperitoneal route 25 mg/kg/day for 4 days. The lesion size footpad was measured four times before infection and treatment, after treatment and at the end of the experiment (days 0, 35, 50, and 100) (Figure 2 and Figure 3). The individual lesion size was calculated from two measurements (differences between the left and the right footpad).

Moreover, with the aim to establish the visceralization of the chronic infection disease, all mice were sacrificed at the end of the experiment to determine parasite burden in spleen by culture on microtiter plates. This in vivo assay indicated that from the day of infection to the day before treatment with (*E*)-piplartine, the progress of the lesion size was similar in the three mice groups (day 35), whereas after the end of treatment (day 50), the mean progress of lesions within groups treated with (*E*)-piplartine and glucantime were reduced by around 35% with respect to the untreated group. At the end of the experiment, both treated groups showed more than 40% reduction in the lesion size and 55% in spleen parasite burden compared to untreated mice group. The intralesion (*E*)-piplartine treatment efficacy was comparable to the intraperitoneal glucantime treatment, with *p*-values of 0.800 and 0.832 for the lesion size and spleen parasite burden, respectively, and significantly higher than the untreated control group, with *p-*values of 0.045 and 0.027, respectively (Table 3).

Thus, the results indicated that (*E*)-piplartine was effective in the in vivo assay (Figure 2 and Figure 3, and Table 3) as evidenced by a significant reduction in the lesion size footpad after infection, and in the spleen parasite burden at the end of the experiment (day 100 post-infection). These findings, together with previous safety [22,25] and pharmacokinetic studies [10,26], provide additional experimental evidence of the potential of (*E*)-piplartine as a promising leishmanicidal lead compound.

In this study, the intralesional route for (*E*)-piplartine was chosen in order to develop a prospective formulation for topical administration. This administration route is an attractive alternative for CL, offering significant advantages over systemic therapy with: fewer adverse side effects, easy administration, and low costs [27]. This later point is relevant as in regions with limited resources there are no dispensaries or qualified personnel for intramuscular or intravenous drug administration [28]. In addition, topical formulations can penetrate over the skin to diminish disease progression at the beginning of the infection [29].

Although the mechanism of action of (*E*)-piplartine has not been established on *Leishmania* parasites, previous studies performed on cancer cell lines [12,30,31] have demonstrated that this alkamide is able to inhibit the proliferative process by activation of mitochondrial apoptosis pathways and induction of reactive oxygen species. Considering these studies, the effect of (*E*)-piplartine on *Leishmania* parasites could also be related to the activation of apoptotic events. Moreover, further studies should be undertaken in order to determine the leishmanicidal mechanism of action of this promising lead compound.

## 4. Conclusions

The results reported herein reinforce the efficacy of (*E*)-piplartine against neglected tropical diseases caused by *Leishmania* spp., and deserve future investigations for further lead optimization with desired drug-likeness properties for the treatment of leishmaniasis. Furthermore, (*E*)-piplartine is a natural alkamide occurring in several species in the widely distributed *Piper* genus, and therefore, the in vivo studies results support and may validate the traditional uses of some *Piper* species by the indigenous people to treat the symptoms of cutaneous leishmaniasis.

## Figures and Tables

**Figure 1 foods-09-01250-f001:**
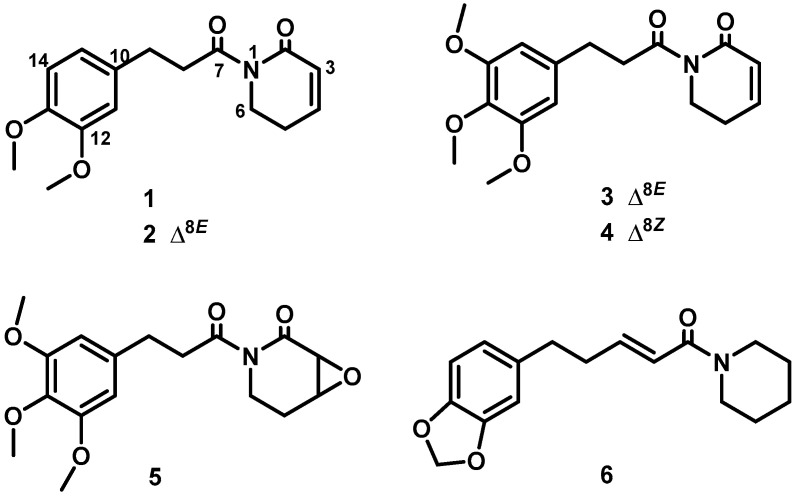
Chemical structures of piperamides (**1**–**6**) isolated from *Piper pseudoarboreum.*

**Figure 2 foods-09-01250-f002:**
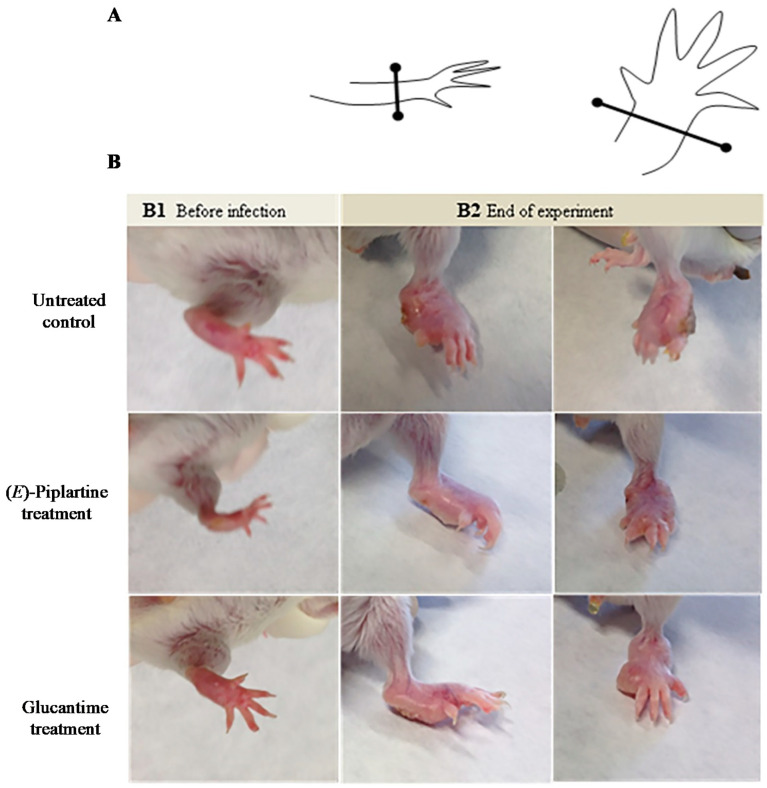
Effect of the treatment with (*E*)-piplartine on chronic cutaneous leishmaniasis in BALB/c mice. (**A**) Graphic representation of footpad measurement: thickness (**left**) and width (**right**). Lesion size is expressed as the difference in size between the infected and contralateral non-infected footpads. (**B**) Representative images of a mouse infected with *Leishmania amazonensis*, at day 0 (**B1**) and at the end of experiment (**B2**).

**Figure 3 foods-09-01250-f003:**
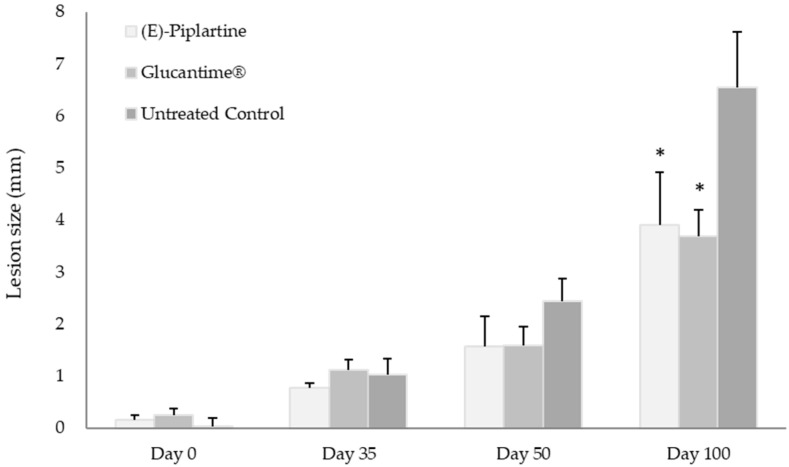
Effectiveness of (*E*)-piplartine in the treatment of chronic cutaneous leishmaniasis. Lesion size was measured four times (day 0, day 35, day 50, and day 100) and expressed as a mean of the group in mm. At end of the experiment, lesion size of (*E*)-piplartine vs. glucantime treated mice groups did not show statistically significant differences with *p-*value > 0.800; untreated control group vs. (*E*)-piplartine and glucantime showed significant differences with *p-*value < 0.000 *.

**Table 1 foods-09-01250-t001:** Leishmanicidal activity on promastigotes forms and cytotoxic activity on macrophages of the extract, fractions, sub-fractions, and isolated compounds *^a^* from *P. pseudoarboreum.*

Sample	*L. amazonensis*		*L. braziliensis*		*L. guyanensis*		*L. infantum*		J774
	IC_50_ *^b^* ± SD	SI *^d^*	IC_50_ *^b^* ± SD	SI *^d^*	IC_50_ *^b^* ± SD	SI *^d^*	IC_50_ *^b^* ± SD	SI *^d^*	CC_50_ *^c^* ± SD
**EtOH**	31.4 ± 2.5	1.8	21.3 ± 1.0	2.6	41.3 ± 1.4	1.3	32.3 ± 1.4	1.7	55.0 ± 4.1
**DCM**	17.7 ± 0.3	1.9	14.7 ± 0.8	2.3	19.1 ± 0.2	1.8	18.4 ± 0.3	1.9	34.1 ± 3.4
**F5**	18.5 ± 0.1	1.2	15.7 ± 2.8	1.4	20.8 ± 0.8	1.1	20.3 ± 0.3	1.1	22.6 ± 2.2
**F6**	2.5 ± 0.0	2.6	2.2 ± 0.1	3.0	3.4 ± 0.2	1.9	3.0 ± 0.1	2.2	6.5 ± 1.3
**F7**	40.0 ± 1.1	0.8	46.7 ± 5.1	0.7	-	-	-	-	30.8 ± 2.8
**1**	28.0 ± 1.0	0.8	28.7 ± 1.0	0.8	24.6 ± 0.0	0.9	27.7 ± 1.7	0.8	23.2 ± 3.1
**2**	1.7 ± 0.3	6.6	1.7 ± 0.0	6.6	2.1 ± 0.0	5.5	3.8 ± 0.3	3.0	11.5 ± 0.7
**3**	2.2 ± 0.0	4.7	1.6 ± 0.3	6.6	2.2 ± 0.3	4.7	2.2 ± 1.6	4.7	10.4 ± 0.9
**4**	3.5 ± 0.0	3.6	2.5 ± 0.0	5.0	3.8 ± 0.0	3.3	5.4 ± 0.3	2.4	12.6 ± 3.8
**5**	-	-	-	-	88.0 ± 2.4	0.6	-	-	52.2 ± 23.0
**M** *^e^*	30.7 ± 0.9	4.4	17.7 ± 0.4	7.7	19.4 ± 1.2	7.0	17.7 ± 1.8	7.7	135.9 ± 10.3

*^a^* Fractions and compounds not included in the table were inactive (IC_50_ > 50 µg/mL and IC_50_ > 100 µM, respectively). *^b^* IC_50_: concentration able to inhibit 50% of parasites. The IC_50_ values of the ethanol extract (EtOH) and dichloromethane (DCM) and F5–F7 fractions are expressed as µg/mL ± standard deviation. The IC_50_ values of the compounds are expressed as µM ± standard deviation. *^c^* CC_50_ concentration able to inhibit 50% of murine macrophages. *^d^* SI: selectivity index (CC_50_/IC_50_). *^e^* M: miltefosine was used as a positive control.

**Table 2 foods-09-01250-t002:** Leishmanicidal activity on amastigote forms of alkamides **2** and **3**.

Compounds	*L. amazonesis*		*L. infantum*	
	IC_50_ *^a^* ± SD	SI *^b^*	IC_50_ ± SD	SI *^b^*
**2**	9.1 ± 0.2	1.3	17.1 ± 0.1	0.7
**3**	8.2 ± 0.1	1.3	16.1 ± 0.1	0.7
**M** *^c^*	49.3 ± 0.2	2.8	23.6 ± 0.4	5.8

*^a^* IC_50_: concentrations able to inhibit 50% of the parasites, and values are expressed as µM ± standard deviation (SD). *^b^* SI: selectivity index (CC_50_ of murine macrophages/IC_50_). *^c^* M: Miltefosine was used as a positive control.

**Table 3 foods-09-01250-t003:** Efficacy of (*E*)-piplartine in the control of the visceralization in chronic Cutaneous Leishmaniasis.

Mice Groups	Spleen Parasite Burden *^a^*		
	Number and Percentage	SEM *^b^* and Percentage	*p*-Value with Untreated Control
Untreated control	277.17 (100%)	±43.3 (15.6%)	-
(*E*)-Piplartine treated	122.6 (44.2%)	±40.3 (14.5%)	0.045
Glucantime^®^ treated	111.6 (40.3%)	±46.8 (16.9%)	0.027

*^a^* Number of parasites per gram in spleen measured on day 100 of infection. *^b^* SEM: Standard error of the mean.

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
