# Peer review of "(E)-Piplartine Isolated from Piper pseudoarboreum, a Lead Compound against Leishmaniasis"

_foods, 2020, doi:10.3390/foods9091250_

Round 1

Reviewer 1 Report

This is an interesting manuscript.  The authors may like to consider the following comments.

Line 38 What does obligating mean? I would remove this word.  Leishmaniasis is caused by infection with Leishmania parasites.

Line 43 - Leishmaniasis cutaneous – this should be ‘Cutaneous leishmaniasis’ – please look at the abbreviation that is conventionally used for this form of leishmaniasis.

Line 47 Do the authors mean ‘American countries’ or South American countries?

Line 50 Do the authors mean adverse side effects?

Line 184.  What volume was injected in treatments? Please state. This is important for readers and review of the methods used by researchers.

Line 207 How were differences between individual treatment determined when a significant difference between treatments was identified using an ANOVA assay? Multiple Mann Whitney assays are inappropriate – as this test is only for comparing two treatments for data that does not conform to parametric tests.   

Table 1/2 Are any of these values significantly different form controls or each other?

Figure 2 What is the maximum lesion size allowed by animal licence used in this study? I ask because the control lesions look larger than we would be allowed in our studies.  The animals would have had difficulty walking with lesions > 3 mm.  Our CL studies are stopped when lesions are < 3 mm so we would not be allowed to have controls that go up to day 100.   I think the controls should have been sacrificed before day 100.

There is no statistical data shown on the Fig – although it is quoted in the text.

Discussion

Can the author indicate how they see the compound they have identified being given to people? Would it be a topical or oral formulation? How does this fit with the solubility or characteristics of the compound?

Author Response

REVIEWER # 1 (Comments for the Author):

Remark 1: Line 38 What does obligating mean? I would remove this word. Leishmaniasis is caused by infection with Leishmania parasites.

ANSWER: This point has been corrected in the revised version following the reviewer’s suggestion (lines 37-38)

Remark 2: Line 43 - Leishmaniasis cutaneous – this should be ‘Cutaneous leishmaniasis’ – please look at the abbreviation that is conventionally used for this form of leishmaniasis.

ANSWER: Thanks to the reviewer for this comment. This point has been corrected in the revised version (line 42)

Remark 3: Line 47 Do the authors mean ‘American countries’ or South American countries?

ANSWER: We mean “New World”. Cutaneous leishmaniasis is endemic not only to South American but also to countries in Central and North American.
“CL is mainly caused by Leishmania major in the Old World and by L. amazonensis and L. braziliensis in the New World, specifically in Brazil.” (line 46

Remark 4: Line 50 Do the authors mean adverse side effects?

ANSWER: Yes, we refer to adverse side effects. This point has been corrected and “adverse effects” has been changed to “adverse side effects” (line 49)

Remark 5: Line 184. What volume was injected in treatments? Please state. This is important for readers and review of the methods used by researchers.

ANSWER: The volume injected in treatments has been included in the revised version (pg 5, line 188).
“..animals treated with (E)-piplartine received in the foot lesions (intralesion) doses of 25 mg/kg/day for 4 days in a 15 μl volume of phosphate saline dilution/propylene glycol ( 9:1)”.

Remark 6: Line 207 How were differences between individual treatment determined when a significant difference between treatments was identified using an ANOVA assay? Multiple Mann Whitney assays are inappropriate – as this test is only for comparing two treatments for data that does not conform to parametric tests.

ANSWER: We thank the reviewer for these comments. We have applied an ANOVA with Tukey post-test to estimate significant differences. This methodology is now included in the Section 2.6 in the revised manuscript.

Remark 7: Table 1/2 Are any of these values significantly different form controls or each other?

ANSWER: Tables 1-2 include the antileishmanial activity of compounds using IC50 on the promastigotes and amastigotes assays. The IC50 is defined as the concentration of drug required to inhibit parasite growth by 50%. To determine the IC50 and CC50, first, the value of the signal-to-noise is subtracted from all the obtained values, then the growth inhibition percentages are calculated for different concentrations of the compound and that of the growth control. Finally, the IC50 and CC50 values are determined by Probit analysis of the values obtained from the different concentrations and the growth control. Single data expressed as IC50 and CC50 do not allow us to estimate significant differences between the compounds.

Remark 8: Figure 2 What is the maximum lesion size allowed by animal licence used in this study? I ask because the control lesions look larger than we would be allowed in our studies. The animals would have had difficulty walking with lesions > 3 mm. Our CL studies are stopped when lesions are < 3 mm so we would not be allowed to have controls that go up to day 100. I think the controls should have been sacrificed before day 100.

ANSWER: In the in-vivo assay, the lesion size evolution was measured periodically, measurements were made of the thickness and width (side by side) of the inflammation of the right (uninfected) and left (infected) paws as shown in Figure 2. This side by side measurement allowed us to reduce errors in the estimation that would be given by measurement of a single side, since the values obtained from inflammation could appear to have a greater volume of inflammation, which has happened in other studies where only the thickness of the paw is measured. In addition, the time of infection of the mice was adapted from that used in similar works to this study. (Cabrera-Serra MG, Valladares B, Piñero JE. In vivo activity of perifosine against Leishmania amazonensis. Acta Trop. 2008,108:20-5. doi: 10.1016/j.actatropica.2008.08.005; Rodrigues RF, Charret KS, Campos MC, Amaral V, Echevarria A, Dos Reis C, Canto-Cavalheiro MM, Leon LL. The in vivo activity of 1,3,4-thiadiazolium-2-aminide compounds in the treatment of cutaneous and visceral leishmaniasis. J. Antimicrob Chemother. 2012, 67:182-90. doi: 10.1093/jac/dkr409)

Remark 9: There is no statistical data shown on the Fig – although it is quoted in the text.

ANSWER: Statistical data have been included in the legend of Figure 3 in the revised manuscript.

Remark 10: Can the author indicate how they see the compound they have identified being given to people? Would it be a topical or oral formulation? How does this fit with the solubility or characteristics of the compound?

ANSWER: Currently, therapeutic failures and the emergence of resistance to conventional drugs are a major problem in the treatment of leishmaniasis. For this reason, combined therapy is the subject of several studies.
Our study provides additional experimental evidences of the potential of (E)-piplartine as a lead candidate for CL, with the aim of finding a drug for topical administration. It could be directly applied to the injury itself and combined with conventional treatments, in order to reduce therapeutic failures, avoid the generation of resistance, and reduce treatment time and minimize side effects. However, studies should be conducted in the
future to improve the pharmacological profile and to develop a formulation (cream, gel or ointment) to incorporate plipartine.
Currently, only a few studies have been focused on the preformulation and/or formulation of piplartine. A paper by Aodah et al reported preformulation studies to determine factors related to solubility and stability which, in turn, can be used to assist future formulation development (PLoS ONE 11: e0151707, 2016). Furthermore, Fofaria et al. published their findings for the preparation of nanoemulsion formulations to formulate piplartine into a suitable drug delivery system for oral drug administration (International Journal of Pharmaceutics 498:12-22, 2016).

I paragraph and some references regarding this point have been added to the Results and Discussion section in the revised manuscript (line 317).
“In this study, the intralesional route for (E)-piplartine was chosen in order to develop a prospective formulation for topical administration. This administration route is an attractive alternative for CL, offering significant advantages over systemic therapy with: fewer adverse side effects, easy administration, and low costs [28]. This later point is relevant as in regions with limited resources there are no dispensaries or qualified personnel for intramuscular or intravenous drug administration [29]. In addition, topical formulations can penetrate over the skin to diminish disease progression at the beginning of the infection [30].” (line 317)

Reviewer 2 Report

OVERALL COMMENTS

The report by Ticona et al. describes the anti-Leishmania activity of six alkamides extracted from the leaves of Piper pseudoarboreum. Overall, the manuscript is well structured and written (except for some typos and minor issues, listed below).

However, I do have a major concern regarding this paper: it it does not provide a substantial amount of new information. In fact, among the six alkamides isolated and tested in this work, the one showing the most potent leishmanicidal activity [(E)-piplartine or piperlongumine], had already been reported to be active against Leishmania (Bodiwala et al. 2007, J Nat Med, 61:418; Moreira et al. 2018, Planta Med 84:1141). The literature in this issue is, nevertheless controversial, with this compound failing to reveal anti-Leishmania activity in other laboratorial contexts (Capello et al. 2015, Nat Prod Commun 10:285; Araújo-Vilges et al. 2017, Pharm Biol 55:1601). Along the manuscript, Ticona et al. put most of these studies into perspective – they, nevertheless skip the work by Capello et al. (2015, Nat Prod Commun 10:285).

Perhaps, the major novelties of the manuscript Ticona et al. are that i) authors carry out a Leishmania inter-species comparative analysis of sensitivity towards (E)-piplartine and the other alkalkines, and ii) this work extends the anti-Leishmania bioassays of (E)-piplartine to an in vivo model of cutaneous leishmaniases. In this last regard, Ticona et al. provide evidence that (E)-piplartine has promising activity as a lead compound to control Leishmania skin infections in murines. Again, this is not the first report testing the leishmanicidal potency of piplartine in a mouse model – Bodiwala et al. (2007, J Nat Med, 61:418) had already reported the efficacy of piplartine in controlling leishmaniases in mice, albeit in this latter case authors assessed visceral infections.

MAJOR COMMENTS

I have some concerns regarding the in vivo assays. Authors compare the leishmanicidal activity of (E)-piplartine with that of the reference drug glucantime, however the regimens that they employed to administer these compounds to mice were not the same – (E)-piplartine was administered directly into skin lesions, whereas glucantime was provided to mice by the intraperitoneal route, meaning that the actual concentration of this drug reaching parasites in the lesions is necessarily below that of (E)-piplartine. In this sense, the comparison made by the authors is somewhat misleading. Authors should comment on this in their manuscript.

At some point (lines 270-272), authors justify progressing with (E)-piplartine to in vivo assays based, among other parameters, the selectivity index (SI) of this compound. This is, nevertheless, a misleading explanation. Indeed, if authors calculate the SI of (E)-piplartine based on the IC50 against the clinically-relevant amastigote form of Leishmania (instead of the insect promastigote stage, as shown in Table 1), they will come to values very close to 1. The same rationale applied to miltefosine reveals SI values in the 2.8-5.6 range – far better than those obtained for (E)-piplartine. SI values calculated based on amastigotes should be included in Table 2, and, importantly, should not be used to justify progressing with (E)-piplartine into vivo assays. Authors can argue that, despite the poor SI values, they have moved on to tests in murines because i) there were no previous indications that this compound is toxic to mice (is this really so?) and also ii) they expected the sensitity of murine-resident amastigotes to (E)-piplartine to be higher than that of the in vitro-cultured amastigotes.

One last note regarding the paragraph starting in line 299. Authors mention that “from the day of infection to the day before treatment with (E)-piplartine, the progress of the lesion size was similar in the three mice groups, whereas 11 days after the end of treatment, the mean progress of lesions within groups treated with (E)-piplartine and glucantime were reduced around 35% respect to the untreated group” without presenting any visual support (be it a Table or Figure).

MINOR COMMENTS

Line 29: Replace “reinforce” with “reinforces”.

Lines 37-38: Replace “Leishmaniasis is a neglected tropical disease caused by several species of obligating intramacrophage protozoan parasites belonging to the genus Leishmania, and is transmitted by the” with “Leishmaniases are a neglected tropical diseases caused by several species of obligatory intramacrophage protozoan parasites belonging to the genus Leishmania, and are transmitted by the”.

Line 39: Replace “Leishmaniasis is” with “Leishmaniases are”.

Line 40: Replace “remains one of the” with “are among the”.

Line 48: Replace “antimoniels pentavalent” with “pentavalent antimonials”.

Line 52: Replace “search” with “searching”.

Line 64: Replace “disease” with “diseases”.

Line 105: Replace “remove” with “removing”.

Line 141: In “in a Schneider's” remove “a”.

Line 162: Replace “by triplicate” with “in triplicate”.

Line 189: In “(intralesion) a doses” remove “a”.

Line 191: Authors refer to “time post-infection” in “weeks”. However, in Figure 3 and along the rest of the document they change to “days”. Please, use a consistent annotation of time along the manuscript.

Line 194: In “The mice” remove “The”.

Line 197: Replace “On 7 day post-incubation” with “On day 7 post-incubation”.

Line 225: Replace “although showed slightly lower selectivity index on macrophages” with “although it showed a slightly low selectivity index taking J774 macrophages as reference mammalian cells”.

Lines 253-254: Where authors write “We assume that the differences in potency depend to a great extend on the infecting Leishmania strain used in the assay”, they should also mentioned that cell culture medium (including the batch of FBS) can also impact of parasites’ response to drugs.

Line 262: Replace “selectivity” with “toxicity”.

Line 265: Replace “Leismania” with “Leishmania”.

Line 319: Replace “revealed” with “reinforced”.

Author Response

REVIEWER # 2 (Comments for the Author):

Remark 1: However, I do have a major concern regarding this paper: it it does not provide a substantial amount of new information. In fact, among the six alkamides isolated and tested in this work, the one showing the most potent leishmanicidal activity [(E)-piplartine or piperlongumine], had already been reported to be active against Leishmania (Bodiwala et al. 2007, J Nat Med, 61:418; Moreira et al. 2018, Planta Med 84:1141). The literature in this issue is, nevertheless controversial, with this compound failing to reveal anti-Leishmania activity in other laboratorial contexts (Capello et al. 2015, Nat Prod Commun 10:285; Araújo-Vilges et al. 2017, Pharm Biol 55:1601). Along the manuscript, Ticona et al. put most of these studies into perspective – they, nevertheless skip the work by Capello et al. (2015, Nat Prod Commun 10:285).

Perhaps, the major novelties of the manuscript Ticona et al. are that i) authors carry out a Leishmania inter-species comparative analysis of sensitivity towards (E)-piplartine and the other alkalkines, and ii) this work extends the anti-Leishmania bioassays of (E)-piplartine to an in vivo model of cutaneous leishmaniases. In this last regard, Ticona et al. provide evidence that (E)-piplartine has promising activity as a lead compound to control Leishmania skin infections in murines. Again, this is not the first report testing the leishmanicidal potency of piplartine in a mouse model – Bodiwala et al. (2007, J Nat Med, 61:418) had already reported the efficacy of piplartine in controlling leishmaniases in mice, albeit in this latter case authors assessed visceral infections.

ANSWER: We agree with the reviewer regarding previous works reporting in vitro [22, 25] and in vivo [24] evaluation of (E)-piplartine against Leishmania spp. Furthermore, Araújo-Vilges and co-workers reported that (E)-piplartine was able to reduce the growth of L. amazonensis promastigotes (MHOM/BR/pH8) in a dose-dependent pattern, exhibiting an IC50 value of 179.0 μg/mL [22]. By contrast, the studies by Capello et al. conclude that (E)-piplartine does not show antileishmanial activity on macrophages infected with L. (L.) amazonensis at 50 μg/ml. Despite controversy regarding the leishmanicidal effect of (E)-piplartine, our in vitro as well as in vivo results support the efficacy of (E)-piplartine providing additional experimental evidences of its potential as a promising leishmanicidal lead compound.
As suggested by the reviewer, a reference by Capello et al has been included in the text as well as in the reference list (reference 23).
Regarding the novelty of the work, as the reviewer pointed out, we carry out a Leishmania inter-species comparative analysis of sensitivity towards (E)-piplartine and other alkamides. Moreover, this study is the first report testing the leishmanicidal potency of (E)-piplartine in an in vivo model of CL. This disease is endemic in more than 70 countries, and has an estimated annual incidence of 1.5-2 million new cases.
The following phrase has been included in the revised version of the manuscript:
“On the other hand, Capello et al. (23) concluded that no antileishmanial activity on macrophages infected with L. (L.) amazonensis was found for (E)-piplartine at 50 μg/ml.” (line 250)

Remark 2. I have some concerns regarding the in vivo assays. Authors compare the leishmanicidal activity of (E)-piplartine with that of the reference drug glucantime, however the regimens that they employed to administer these compounds to mice were not the same – (E)-piplartine was administered directly into skin lesions, whereas
glucantime was provided to mice by the intraperitoneal route, meaning that the actual concentration of this drug reaching parasites in the lesions is necessarily below that of (E)-piplartine. In this sense, the comparison made by the authors is somewhat misleading. Authors should comment on this in their manuscript.

ANSWER: In the therapeutics of cutaneous or mucosal leishmaniasis, the administration of Glucantime® are by intramuscular or intravenous routes. The intraperitoneal route is equivalent to the intravenous route, therefore, we consider that it was more suitable for administration in mice and also avoids variations due to the absorption process.
The following paragraph and some references have been added in the revised manuscript.
“In this study, the intralesional route for (E)-piplartine was chosen in order to develop a prospective formulation for topical administration. This administration route is an attractive alternative for CL, offering significant advantages over systemic therapy with: fewer adverse side effects, easy administration, and low costs [28]. This later point is relevant as in regions with limited resources there are no dispensaries or qualified personnel for intramuscular or intravenous drug administration [29]. In addition, topical formulations can penetrate over the skin to diminish disease progression at the beginning of the infection [30].” (line 317)

Remark 3. At some point (lines 270-272), authors justify progressing with (E)-piplartine to in vivo assays based, among other parameters, the selectivity index (SI) of this compound. This is, nevertheless, a misleading explanation. Indeed, if authors calculate the SI of (E)-piplartine based on the IC50 against the clinically-relevant amastigote form of Leishmania (instead of the insect promastigote stage, as shown in Table 1), they will come to values very close to 1. The same rationale applied to miltefosine reveals SI values in the 2.8-5.6 range – far better than those obtained for (E)-piplartine. SI values calculated based on amastigotes should be included in Table 2, and, importantly, should not be used to justify progressing with (E)-piplartine into vivo assays. Authors can argue that, despite the poor SI values, they have moved on to tests in murines because i) there were no previous indications that this compound is toxic to mice (is this really so?) and also ii) they expected the sensitity of murine-resident amastigotes to (E)-piplartine to be higher than that of the in vitro-cultured amastigotes.

ANSWER: Thanks to the reviewer for the above comments. As the reviewer suggests, SI values calculated based on amastigotes have been included in Table 2 in the revised manuscript.
Furthermore, in the revised manuscript we have included some comments regarding this point.
“Taking into consideration its potency and efficacy on Leishmania promastigote and amastigote forms, and although a poor selectivity index, (E)-piplartine was selected for in vivo assays to investigate its potential as a lead compound targeting CL since previous toxicological studies indicate a good safety profile in murine models [24]” (line 270)

Remark 4. One last note regarding the paragraph starting in line 299. Authors mention that “from the day of infection to the day before treatment with (E)-piplartine, the progress of the lesion size was similar in the three mice groups, whereas 11 days after the end of treatment, the mean progress of lesions within groups treated with (E)-
piplartine and glucantime were reduced around 35% respect to the untreated group” without presenting any visual support (be it a Table or Figure).

ANSWER: Thank you for these comments. This paragraph has been corrected in the revised version
"This in vivo assay indicated that from the day of infection to the day before treatment with (E)-piplartine, the progress of the lesion size was similar in the three mice groups (day 35), whereas after the end of treatment (day 50), the mean progress of lesions within groups treated with (E)-piplartine and glucantime were reduced by around 35% with respect to the untreated group." (line 303)

Remark 5. MINOR COMMENTS

All the reviewer’s suggestions have been addressed in the revised version.

Line 191: Authors refer to “time post-infection” in “weeks”. However, in Figure 3 and along the rest of the document they change to “days”. Please, use a consistent annotation of time along the manuscript.

ANSWER: This point has been corrected in the revise manuscript.
“The measurement of cutaneous lesion was monitored at 0, 35, 50 and 100 days post-infection” (line 192)

Lines 253-254: Where authors write “We assume that the differences in potency depend to a great extend on the infecting Leishmania strain used in the assay”, they should also mentioned that cell culture medium (including the batch of FBS) can also impact of parasites’ response to drugs.

ANSWER: As the reviewer has suggested, we have been corrected this point.
“We assume that the differences in potency depend to a great extent on the infecting Leishmania strain used in the assay and on cell culture procedures”. (line 251)

Reviewer 3 Report

The authors isolated six compounds from Piper pseudoarboreum. They used in vitro screening to select (E)-piprastin as an effective compound for in vivo research. (E)-Piplartine showed efficacy against Lactobacillus amazonis infection. This work is very interesting for medicinal chemists. This work is very interesting for medicinal chemists. The work was well presented. However, at this moment I cannot recommend publication without the original data. 

I strongly suggest that the authors provide characterization data and spectra of the isolated compounds, and the original data of the in vitro screening and in vivo study. 

  1. The authors claimed that "Their chemical structures were elucidated on the basis of their spectrometric and spectroscopic data, including mono- (1H and 13C) and bi-dimensional (COSY, HSQC, and HMBC) NMR experiments, and comparison with data reported in the literature." However, I didn't see any data supporting their claim. Please provide 1H NMR analysis data including coupling values along with the spectra as supplemental files. Provide 13C NMR analysis data along with the spectra as supplemental files. COSY, HSQC, and HMBC data along with the spectra. 
  2. Figure 1. Atom labels need to be regulated. Please keep structures in a consistent format. 
  3. In-vitro results, please use * indicate the p-values in all statistic data. 
  4. In the experimental section, the authors stated that "The efficacy of each compound was estimated by calculating the IC50 (concentration of the compound that produced a 50% reduction in parasites) and GI% (percentage of growth inhibition)." However, there is only provided IC50 which requires the original statistic results of each group and each concentration. Please attach the original data as supplemental files for the evaluation of In-vitro Promastigotes Susceptibility Assay, Cytotoxicity Assay, and In-vitro Amastigote Assay. 
  5. The in-vivo study has 3 randomly separated mice groups (8 mice per group). The authors stated that "The lesion size footpad was measured four times before infection and treatment, after treatment and at the end of the experiment (days 0, 35, 50 and 100)". However, I didn't find the data. Please provide the original data of the lesion size footpad of each mouse (days 0, 35, 50, and 100).
  6. The results and discussion section is more like the description of the results. Rarely see discussion. For example, interpretation of the reason why the compounds better than Miltefosine positive group. What is the mechanism behind these results? 

Author Response

REVIEWER # 3 (Comments for the Author):

Remark 1. The authors claimed that "Their chemical structures were elucidated on the basis of their spectrometric and spectroscopic data, including mono- (1H and 13C) and bi-dimensional (COSY, HSQC, and HMBC) NMR experiments, and comparison with data reported in the literature." However, I didn't see any data supporting their claim. Please provide 1H NMR analysis data including coupling values along with the spectra as supplemental files. Provide 13C NMR analysis data along with the spectra as supplemental files. COSY, HSQC, and HMBC data along with the spectra.

ANSWER: Regarding this point, I would like to highlight, as an expert on natural products chemistry, that it is usual to include all the spectroscopic data and spectra for compounds that have not been previously reported, but not for known compounds. The known compounds are usually documented with a reference in which all these data are reported for the first time.
Despite this and, as suggested by the reviewer, in the revised manuscript a SI with the 1H NMR and 13C NMR spectra of the isolated compounds (1-6), in addition to the reference for each compound included in the manuscript, has now been included.

Remark 2. Figure 1. Atom labels need to be regulated. Please keep structures in a consistent format.

ANSWER: I checked Figure 1 and Atom labels are all in Arial 9, whereas the numbers in the numbering structure (compound 1) are in Arial 7. This is the usual way to draw structures, setting the numbered carbons smaller than the atom labels, otherwise they look too big in comparison with the structure.

Remark 3. In-vitro results, please use * indicate the p-values in all statistic data.

ANSWER: This point has been corrected in Figure 2.

Remark 4. In the experimental section, the authors stated that "The efficacy of each compound was estimated by calculating the IC50 (concentration of the compound that produced a 50% reduction in parasites) and GI% (percentage of growth inhibition)." However, there is only provided IC50 which requires the original statistic results of each group and each concentration. Please attach the original data as supplemental files for the evaluation of In-vitro Promastigotes Susceptibility Assay, Cytotoxicity Assay, and In-vitro Amastigote Assay.

ANSWER: This point has been corrected in Material and Methods (Section 2.5.4).

Remark 5. The in-vivo study has 3 randomly separated mice groups (8 mice per group). The authors stated that "The lesion size footpad was measured four times before infection and treatment, after treatment and at the end of the experiment (days 0, 35, 50 and 100)". However, I didn't find the data. Please provide the original data of the lesion size footpad of each mouse (days 0, 35, 50, and 100).

ANSWER: Results of the in vivo assay (effect of the treatment with (E)-piplartine on CL in BALB/c mice) and lesion size footpad measurements are included in Figures 2 and 3.
In addition, original data of the in vitro and in vivo studies have now been included in Supporting Materials for review only.

Remark 6. The results and discussion section is more like the description of the results. Rarely see discussion. For example, interpretation of the reason why the compounds better than Miltefosine positive group. What is the mechanism behind these results?

ANSWER: As suggested by the reviewer, a discussion and some references regarding the mechanism of action of piplartine have been included in the revised manuscript

“Although the mechanism of action of (E)-piplartine has not been established on Leishmania parasites, previous studies performed on cancer cell lines [12, 31, 32] have demonstrated that this alkamide is able to inhibit the proliferative process by activation of mitochondrial apoptosis pathways and induction of reactive oxygen species. Considering these studies, the effect of (E)-piplartine on Leishmania parasites could also be related to the activation of apoptotic events. Moreover, further studies should be undertaken in order to determine the leishmanicidal mechanism of action of this promising lead compound” (line 324)

Round 2

Reviewer 3 Report

ANSWER: Regarding this point, I would like to highlight, as an expert on natural products chemistry, that it is usual to include all the spectroscopic data and spectra for compounds that have not been previously reported, but not for known compounds. The known compounds are usually documented with a reference in which all these data are reported for the first time.

I agree that known compounds generally do not need supporting spectra, but at least the authors need to provide a summary of NMR data or spectra to reviewers. Otherwise, how does the reviewer know if the authors have isolated the same compounds as the reference? Just rely on the authors' claim? In addition, I believe the authors hope that more scientists in different fields will read their report, not limited to natural product chemists.

Remark 3. In-vitro results, please use * indicate the p-values in all statistic data.
ANSWER: This point has been corrected in Figure 2.

What does this mean? corrected in Figure 2, p-values? 

BTW Figure 3 caption showed p-value < 0.000* and p-value < 0.000**. please correct it. 

Author Response

REVIEWER # 3 (Comments for the Author):

Remark 3. In-vitro results, please use * indicate the p-values in all statistic data.

ANSWER: This point has been corrected in Figure 2.

What does this mean? corrected in Figure 2, p-values?

ANSWER: The review is right as the previous answer to remark 3 was wrong.
In vitro assays determinate the antileishmanial activity of the compounds by calculating IC50s on promastigote and amastigote forms. The IC50 is defined as the concentration of drug required to inhibit parasite growth by 50%. To determine IC50s and CC50s, first, the value of the signal-to-noise is subtracted from all the obtained values, and after that the growth inhibition percentages are calculated for different concentrations of the compound and that of the growth control. Finally, the IC50 and CC50 are determined by Probit analysis between the values obtained from the different concentrations and the growth control. Single data expressed as IC50 and CC50 does not allow us to estimate significant differences between the compounds. Statistical analyzes are no applicable in this case. (Sebaugh JL. Guidelines for accurate EC50/IC50 estimation. Pharm Stat. 2011; 10(2):128-134. doi:10.1002/pst.426)

Remark. BTW Figure 3 caption showed p-value < 0.000* and p-value < 0.000**. please correct it.

ANSWER. Figure 3 caption has been corrected in the revised manuscript in accordance with the reviewer’s suggestion.